# Bounds on the Generalization Error in Active Learning

Vincent Menden[1], Yahya Saleh[1], and Armin Iske[1]

[1]Department of Mathematics, Universität Hamburg, Bundesstr. 55, 20146, Hamburg, Germany
{vincent.menden}@studium.uni-hamburg.de
{yahya.saleh, armin.iske}@uni-hamburg.de

## Abstract

We establish empirical risk minimization principles for active learning by deriving a family of upper bounds on the generalization error. Aligning with empirical observations, the bounds suggest that superior query algorithms can be obtained by combining both informativeness and representativeness query strategies, where the latter is assessed using integral probability metrics. To facilitate the use of these bounds in application, we systematically link diverse active learning scenarios, characterized by their loss functions and hypothesis classes to their corresponding upper bounds. Our results show that regularization techniques used to constraint the complexity of various hypothesis classes are sufficient conditions to ensure the validity of the bounds. The present work enables principled construction and empirical quality-evaluation of query algorithms in active learning.

## 1 Introduction

Empirical risk minimization (ERM) principles are at the heart of statistical learning theory. In addition to laying a formal mathematical foundation for supervised-learning algorithms, they lead to substantial advances in algorithmic design, such as the development of max-margin methods [1, 2]. However, the majority of ERM principles considered the standard passive supervised-learning setting, and formal principles for other settings such as online or semi-supervised learning are largely missing.

An important such setting is that of active learning (AL), where, similar to the standard supervised-learning setting, computer oracles learn a probability distribution that models a certain phenomenon given a finite set of observations. However, unlike in the standard passive-learning setting, the oracle in AL also selects an optimal, minimal set of observations to achieve this goal. Even in the age of big data, numerous applications require this setting, mainly due to high computational costs corresponding to the annotation, i.e., labeling of datapoints [3]. For example, in the emerging field of physics-informed neural networks, it is often required to learn solutions or solution operators of high-dimensional partial differential equations [4, 5]. Generating the training data in such learning tasks involve running computationally expensive numerical solvers. AL is, indeed, a very appealing setting for such problems and has been extensively applied for, e.g., parameteric Schrödinger equations [6–8].

The crucial task in all AL scenarios is to *query* the labels of the most useful datapoints while minimizing the number of queries [3]. The rationale behind the design of such *query algorithms* can be divided into two categories [3, 9]. The first category relies on the *informativeness* criterion [10, 11], where the query algorithm aims at selecting the most informative samples, whereby shrinking the space of the candidate hypotheses as fast as possible. Such query algorithms indeed introduce a sampling bias [9], as the selected training dataset is not necessarily i.i.d. sampled from the true distribution. This renders the query algorithm prone to oversampling outliers that are not very representative of the application domain, where the model would be employed [3, 12]. The second category is based on the *representativeness* criterion, where the query algorithm aims at selecting samples that are representative of the patterns present in the unlabeled data. Such methods tend to perform well when only a small labelled dataset is available, but their performance rather deteriorate with increasing labeled-dataset size. Numerous empirical and theoretical studies indeed point out that superior query algorithms can be obtained by combining both criteria [9, 12, 13].

AL algorithms are often heuristic in designing the specific query criterion or ad hoc in measuring and combining the informativeness and representativeness of the samples. For example, a common heuristic to combine both criteria is to query data points by a random-sample selection that gives higher weights to samples corresponding to large uncertainties. Since the selection of new points is random, the query algorithm ends up querying representative datapoints. While such heuristics are often successful in practice, they lack a principled approach and are often domain-specific [3]. Some first steps into a more principled approach to AL were taken in Wang and Ye [14], where the authors derived an upper bound on the generalization error using the maximum mean discrepancy (MMD) as a measure of the representativeness of a sample. Later, a similar result was obtained using the Wasser-

Proceedings of the 6th Northern Lights Deep Learning Conference (NLDL), PMLR 265, 2025.

stein distance as a measure of representativeness [15]. However, these results assumed rather harsh conditions on the loss function and the supervised-learning problem that restrict the applicability of these upper bounds.

**Organization.** In Section 2 we cite the ERM principle in passive learning and introduce the notion of integral probability metrics (IPMs). In Subsection 3.1 we establish an ERM principle for AL. In Subsection 3.2 we link the upper bound in the ERM principle to two learning settings, employing linear models with the $\ell_1$-loss function, and deep neural networks with the hinge loss, respectively.

## Notation

On the probability measure space $(\Omega, \mathcal{A}, P)$ we consider the random vector $X : \Omega \to \mathbb{X} \subseteq \mathbb{R}^n$ and the random variable $Y : \Omega \to \mathbb{Y} \subseteq \mathbb{R}$. To simplify the terminology we refer to $X$ by a random variable irrespective of the value of $n$. We set $Z = (X, Y)$ to be the joint random variable and denote by $P_Z$ its probability distribution on $\mathbb{Z} := \mathbb{X} \times \mathbb{Y}$. We denote by $P_X$ the marginal probability distribution and by $P_{Y|X}$ the conditional probability, i.e., $P_Z = P_X P_{Y|X}$. To describe the queried data we introduce the random variable $Q : \Omega \to \mathbb{Q} \subseteq \mathbb{R}^n$ with distribution $P_Q$.

Throughout the paper we denote by $\mathfrak{H}$ a generic hypothesis class containing learners $h : \mathbb{X} \to \mathbb{Y}$ and by $\ell : \mathbb{Y}^2 \to \mathbb{R}_{\geq 0}$ a generic loss function that evaluates the deviation of a prediction $\hat{y} = h(x)$ from the true label $y$. For such a loss function, a fixed $y \in \mathbb{Y}$ and a fixed $h \in \mathfrak{H}$, we define $\ell^y : \mathbb{X} \to \mathbb{R}$ by $\ell^y(x) := \ell(y, h(x))$.

For a fixed $\mathfrak{H}$ and $\ell$ we denote by $R_{\mathcal{P}_Z}(h)$ the true risk of a hypothesis $h \in \mathfrak{H}$ with respect to $P_Z$, i.e.,

$$R_{\mathcal{P}_Z}(h) := \int_{\mathbb{Z}} \ell(y, h(x)) \, dP_Z(x, y).$$

Given a dataset of finite observations $D_m := \{z_1 = (x_1, y_1), \ldots, z_m = (x_m, y_m)\}$, we denote by $\hat{R}(h; D_m)$ the empirical risk of the hypothesis $h$, i.e.,

$$\hat{R}(h; D_m) := \frac{1}{m} \sum_{i=1}^m \ell(y_i, h(x_i)).$$

Additionally, we define

$$\mathcal{K} := \ell \circ \mathfrak{H} \circ D_m$$
$$:= \{\ell(y_i, h(x_i)) : h \in \mathfrak{H}, (x_i, y_i) \in D_m\}.$$

Finally, for a vector $v \in \mathbb{R}^n$ we denote by $\|v\|_2$ the standard 2-norm, i.e., $\|v\|_2 = \sum_{i=1}^n \sqrt{w_i^2}$. Similarly, we set $\|v\|_1 = \sum_{i=1}^n |v_i|$ and for a matrix $M \in \mathbb{R}^{n \times m}$ we consider the spectral-2-norm $\|M\|_2 := \sup_{\|v\|_2 = 1} \|Mv\|_2$. For compact sets $A \subset \mathbb{R}^n$ we set $M_A := \max_{a \in A} \|a\|_2$.

## 2  Preliminaries

In standard supervised learning, the unachievable goal of minimizing the true risk is replaced by minimizing the empirical risk over a finite sample, while imposing constraints on the complexity of the hypothesis class, often using regularization techniques.

Formally, this common practice in supervised learning can be understood as an inductive principle, where the minimization of the true risk is replaced by the minimization of an upper bound to it. Such upper bounds exist in a variety of forms, often involving different notions of complexity of the hypothesis class [1, 2, 16]. As an example, we cite the following celebrated result.

**Theorem 1.** Assume that $\ell(y, h(x)) \leq k$ for some $k > 0$, any $h \in \mathfrak{H}$ and any $(x, y) \in \mathbb{Z}$. Then, for any $\delta > 0$ and any $h \in \mathfrak{H}$, with probability of at least $1 - \delta$ over the choice of the training set $D_m$ it holds that

$$R_{P_Z}(h) \leq \hat{R}_{D_m \sim P_Z}(h) + 2 \operatorname{Rad}(\mathcal{K})$$
$$+ k \sqrt{\frac{2 \log(\frac{4}{\delta})}{m}}, \quad (1)$$

where $\operatorname{Rad}(\mathcal{K})$ is the Rademacher complexity defined by

$$\operatorname{Rad}(\mathcal{K}) := \mathbb{E}_\sigma \left[ \sup_{k \in \mathcal{K}} \frac{1}{m} \sum_{i=1}^m \sigma_i k(x_i) \right],$$

where $\mathbb{E}_\sigma$ denotes the expectation operator with respect to the distribution of $\sigma$.

*Proof.* See Shalev-Shwartz and Ben-David [16, Theorem. 26.5]. $\square$

Minimizing the upper bound in (1) was shown to be equivalent to common supervised-learning practices across a variety of loss functions and hypothesis classes. Moreover, such upper bounds were shown to accommodate novel statistical behaviors, such as the generalization error of deep neural networks [17].

Similar to the standard supervised-learning setting, the goal in AL is to find a hypothesis of $h \in \mathfrak{H}$ that minimizes the true risk. However, to achieve this goal, the oracle in AL is required to select a minimal set of observations. This often violates the passive-learning assumption that the training data is i.i.d. sampled from the true distribution. Generally, the training data $D$ in AL follows the distribution $P_{\hat{Z}} := P_Q P_{Y|X}$, i.e., it shares the same conditional distribution as the true distribution $P_Z$, but has a different marginal distribution $P_Q$. The choice of an optimal query algorithm can, thus, be framed as finding an optimal marginal distribution $P_Q$.

The representativeness criterion in AL can be understood as the requirement that $P_Q$ does not deviate too much from the true marginal $P_X$. To quantify this deviation, we use the notion of IPM [18].

**Definition 1** (Integral Probability Metrics). Consider the measure space $(\mathbb{X}, \mathcal{B}(\mathbb{X}))$ where $\mathcal{B}(\mathbb{X})$ denotes the Borel $\sigma$-algebra generated by $\mathbb{X} \subset \mathbb{R}^n$. Further let $\mathcal{F} \subseteq \mathcal{B}_C$ with $\mathcal{B}_C$ the set of real-valued measurable functions on $\mathbb{X}$, which are bounded by $C > 0$. Then, for two probability measures $P_X$ and $P_Q$ on $(\mathbb{X}, \mathcal{B}(\mathbb{X}))$ we define the integral probability metric with respect to the generator $\mathcal{F}$ as

$$d_{\mathcal{F}}(P_X, P_Q) := \sup_{f \in \mathcal{F}} \left| \int_{\mathbb{X}} f(x) dP_X(x) - \int_{\mathbb{X}} f(q) dP_Q(q) \right| \quad (2)$$

Choosing different generators $\mathcal{F}$ in (2) leads to different statistical distances. We consider the following two generators:

(1) The *Total Variation metric* $(d_{\mathcal{F}_{\mathrm{TV}}})$ is obtained by considering

$$\mathcal{F}_{\mathrm{TV}} := \{ f : \mathbb{X} \to \mathbb{R} : \|f\|_{\infty} \leq 1 \},$$

where $\|f\|_{\infty}$ denotes the supremum norm.

(2) The *Kantorovic metric* $(d_{\mathcal{F}_K})$ is obtained by considering

$$\mathcal{F}_K := \{ f : \mathbb{X} \to \mathbb{R} : \|f\|_L \leq 1 \},$$

where

$$\|f\|_L := \sup \left\{ \frac{|f(x) - f(y)|}{\|x - y\|_2} : x \neq y, x, y \in S \right\}$$

denotes the Lipschitz semi-norm on a metric space $(S, \rho)$.

To establish the ERM principle for AL, we need the following concept.

**Definition 2** (Maximal Generator). Let $\mathcal{F} \subseteq \mathcal{B}_C$ be a generator. We define the set of maximal generators $\mathcal{R}_{\mathcal{F}}$ to be the set of functions $f \in \mathcal{B}_C$ with the property

$$\left| \int_{\mathbb{X}} f(x) dP_X(x) - \int_{\mathbb{X}} f(q) dP_Q(q) \right| \leq d_{\mathcal{F}}(P_X, P_Q),$$

for all probability measures $P_X$ and $P_Q$ on $(\mathbb{X}, \mathcal{B}(\mathbb{X}))$.

In other words, $\mathcal{R}_{\mathcal{F}}$ describes the largest set in $\mathcal{B}_C$ preserving the value of $d_{\mathcal{F}}(\cdot, \cdot)$. It is clear that $\mathcal{F} \subset \mathcal{R}_F$.

**Lemma 1.** Let $(\mathbb{Y}, \mathcal{B}(\mathbb{Y}), P)$ be a probability space, $\mathcal{F} \subset \mathcal{B}_C$ a generator and $f : \mathbb{Y} \times \mathbb{X} \to \mathbb{R}$ a $\mathcal{B}(\mathbb{Y} \times \mathbb{X})$-measurable function with $f(y, \cdot) \in \mathcal{F} \subset \mathcal{B}_C$ for all $y \in \mathbb{Y}$. Then

$$g(\cdot) := \int_{\mathbb{Y}} f(y, \cdot) dP(y)$$

is a well-defined function on $\mathbb{X}$ and it holds that $g \in \mathcal{R}_{\mathcal{F}}$.

*Proof.* See Müller [18, Theorem 3.4]. $\qquad \square$

Note that Lemma 1 also holds for any $f \in \mathcal{R}_{\mathcal{F}}$. The stage is now ready to state our results.

# 3  ERM in Active Learning

We begin by establishing the ERM principle for AL, where the IPM is used as a measure of representativeness.

## 3.1  Bounding the True Risk

We recall that the training data in AL is assumed to follow a distribution $P_{\hat{Z}}$ that shares the same conditional distribution of the generating distribution $P_Z$, i.e., $P_{\hat{Z}} = P_Q P_{Y|X}$. Further, recall that a given a loss function $\ell : \mathbb{Y}^2 \to \mathbb{R}_{\geq 0}$ induces the function $\ell^y : \mathbb{X} \to \mathbb{R}$ by $\ell^y(x) := \ell(y, h(x))$ for some $y \in \mathbb{Y}$ and $h \in \mathfrak{H}$. Lastly, recall that the set $\mathcal{B}_C$ contains all real-valued measurable functions on $\mathbb{X}$, which are bounded by $C > 0$.

**Theorem 2** (ERM principle for AL). Let $\mathcal{F} \subset \mathcal{B}_C$ be a generator for some $C > 0$, and $\ell : \mathbb{Y}^2 \to \mathbb{R}_{\geq 0}$ be a loss function that satisfies the hypothesis of Theorem 1. Further, let $\ell^y \in \mathcal{F}$ for all $y \in \mathbb{Y}$ and $h \in \mathfrak{H}$ and $\hat{D}_m = \{\hat{Z}_1, \ldots \hat{Z}_m\} \sim P_{\hat{Z}}$ be an i.i.d sample. Then, with probability of at least $1 - \delta$ and for any $h \in \mathfrak{H}$, we have

$$R_{P_Z}(h) \leq \hat{R}_{\hat{D}_m \sim P_{\hat{Z}}}(h) + d_{\mathcal{F}}(P_X, P_Q) \\ + 2 \, \mathrm{Rad}(l \circ \mathfrak{H} \circ \hat{D}_m) \\ + k \sqrt{\frac{2 \log(\frac{4}{\delta})}{m}}. \quad (3)$$

*Proof.* We note that the hypothesis of this theorem satisfies the conditions of Theorem 1. Therefore, it follows that

$$R_{P_Z}(h) \leq R_{P_Z}(h) - R_{P_{\hat{Z}}}(h) \\ + \hat{R}_{\hat{D}_m \sim P_{\hat{Z}}}(h) \\ + 2 \, \mathrm{Rad}(l \circ \mathfrak{H} \circ \hat{D}_m) \quad (4) \\ + k \sqrt{\frac{2 \log(\frac{4}{\delta})}{m}}.$$

Set $K(h) := R_{P_Z}(h) - R_{P_{\hat{Z}}}(h)$ and note that

$$K(h) = \int_{\mathbb{X}} \int_{\mathbb{Y}} l(y, h(x)) dP_{Y|X}(y) dP_X(x) \\ - \int_{\mathbb{X}} \int_{\mathbb{Y}} l(y, h(x)) dP_{Y|X}(y) dP_Q(x) \\ = \int_{\mathbb{X}} \int_{\mathbb{Y}} l(y, h(x)) dP_{Y|X}(y) dP_X(x) \\ - \int_{\mathbb{X}} \int_{\mathbb{Y}} l(y, h(x)) dP_{Y|X}(y) dP_Q(x)$$

by virtue of Fubini's theorem. Set

$$g := \int_{\mathbb{Y}} l(y, h(\cdot)) dP_{Y|X}(y) \quad (5)$$

and note that $\ell(y, h(\cdot)) = \ell^y$ satisfies all the hypotheses of Lemma 1 and hence $g \in \mathcal{R}_\mathcal{F}$. Thus, using the definition of $\mathcal{R}_\mathcal{F}$ we can estimate

$$
\begin{aligned}
D(h) &= \int_\mathbb{X} g(x) dP_X(x) - \int_\mathbb{X} g(x) dP_Q(x) \\
&\leq \sup_{f \in \mathcal{R}_\mathcal{F}} \left| \int_\mathbb{X} f(x) dP_X(x) - \int_\mathbb{X} f(x) dP_Q(x) \right| \\
&= \sup_{f \in \mathcal{F}} \left| \int_\mathbb{X} f(x) dP_X(x) - \int_\mathbb{X} f(x) dP_Q(x) \right| \\
&= d_\mathcal{F}(P_X, P_Q).
\end{aligned}
$$

$\square$

**Remark.** We note that an upper bound on the true risk in AL using the IPM appeared in the work of Wang and Ye [14]. However, to derive their result the authors made a direct assumption on $g$, see (5). A more refined version appeared in the work of Saleh [19], where the author derived direct conditions on the loss function $\ell$ that would reduce the IPM to the Kantorovic metric and the MMD. Theorem 2 can be considered as a more general formulation of these results that allow a direct connection to the literature on maximal generators.

Theorem 2 establishes an ERM principle, which is in accordance with common practices in AL. To see this consider a classification task and assume that the AL oracle has access to a hypothesis class $\mathfrak{H}$, an initially labelled dataset $D^{(0)} \sim P_Z$ and a pool of unlabeled data that is i.i.d. sampled from $P_X$. The upper bound suggests finding a hypothesis $h$ and sampling an additional dataset $D^{(1)}$ that minimize the empirical risk. A certain hypothesis $h$ that minimizes the empirical risk on $D^{(0)}$ would benefit the most from a dataset $D^{(1)}$ that is close to the decision boundary. This corresponds to the concept of informativeness sampling in AL. In addition, the upper bound in the theorem suggests that a query strategy should sample points, whose distribution is close to the true marginal distribution of the data. In other words, an optimal query strategy should sample points that are representative of the underlying marginal. Indeed, a balance between these two criteria is crucial for the success of an AL query algorithm [9, 12, 13].

## 3.2 Mapping Learning Settings to Generalization Bounds

The upper bound derived in Theorem 2 is generic and can take many forms by choosing different generators $\mathcal{F}$. We aim in this section at deriving explicit bounds for the true risk given a certain learning setting. Nevertheless, additional restrictions on the learning setting need to be imposed for Theorem 2 to hold. We consider the learning setting to be determined by a choice of the hypothesis class $\mathfrak{H}$, the

domain $\mathbb{X}$, the codomain $\mathbb{Y}$ and the loss function $\ell$. In the following $\mathbb{X} \subset \mathbb{R}^n$ and $\mathbb{Y} \subset \mathbb{R}$ unless otherwise specified.

We consider first a regression task employing the linear hypothesis class

$$
\mathfrak{H}_L := \{h : \mathbb{X} \to \mathbb{Y} : h(x) = w^T x + b, w \in \mathbb{R}^n, b \in \mathbb{R}\},
$$

where $w$ and $b$ are the learnable parameters along with the loss $\ell_1(y, h(x)) := |y - h(x)|$ defined for any $y \in \mathbb{Y}$ and $h \in \mathfrak{H}_L$.

**Theorem 3** (Linear Hypothesis Classes). Consider a regression problem employing $\mathfrak{H}_L$ and the $\ell_1$-loss. Assume that $w$ is such that $\|w\|_2 \leq 1$. Then the true risk of a hypothesis $h \in \mathfrak{H}_L$ can be bounded as in Theorem 2 by choosing the generator $\mathcal{F} = \mathcal{F}_K$.

*Proof.* Analogous to our previous notation we set $\ell_1^y(x) := \ell_1(y, h(x))$ for any $y \in \mathbb{Y}$.

Fix $y \in \mathbb{Y}$ and $h \in \mathfrak{H}_L$. By Theorem 2, it suffices to show that $\ell_1^y \in \mathcal{F}_K$. For any $x_1, x_2 \in \mathbb{X}$, it holds that

$$
\begin{aligned}
|\ell_1^y(x_1) - \ell_1^y(x_2)| &= \left| |h(x_1) - y| - |h(x_2) - y| \right| \\
&\leq \left| (w^T x_1 + b) - (w^T x_2 + b) \right| \\
&\leq \left| w^T(x_1 - x_2) \right| \\
&\leq \|w\|_2 \|x_1 - x_2\|_2,
\end{aligned}
$$

where we used the reversed-triangle inequality and the Cauchy-Schwarz inequality. Setting $\|w\|_2 \leq 1$ implies that $\|\ell_1^y\|_L \leq 1$ and hence $\ell_1 \in \mathcal{F}_K$. $\square$

Theorem 1 suggests that the natural regularization constraint $\|w\|_2 \leq 1$, commonly used for mitigating overfitting, is sufficient to bound the true risk of a linear hypothesis class in an AL setting.

We now look at an example of a binary classification problem, i.e., $\mathbb{Y} = \{-1, 1\}$, using feed-forward neural networks

$$
\mathfrak{H}_{NN} := \{h : \mathbb{X} \to \mathbb{Y} : h(x) = \text{sign}(o^T f(x) + b)\}
$$

with weight $o \in \mathbb{R}^n$ and bias $t \in \mathbb{R}$ in the output layer and the neural network function $f(x) = W^{(L)} \sigma(W^{(L-1)} \cdots \sigma(W^{(1)} x + b^{(1)}) \cdots + b^{(L-1)}) + b^{(L)}$, where $\sigma$ is the ReLU activation function, and $W^{(l)}$, $b^{(l)}$ are the weight matrices and bias vectors, respectively. The learnable parameters are assumed to have arbitrary finite dimensions. We consider the hinge loss $\ell_H(y, h(x)) = \max(0, 1 - y(w^T x + b))$.

**Theorem 4** (Neural Networks). Consider a binary classification task employing $\mathfrak{H}_{NN}$ and the $\ell_H$-loss. Assume that $\|o\|_2 \prod_{i=1}^{L} \|W\|_2 \leq 1$, then the true risk of a hypothesis $h \in \mathfrak{H}_{NN}$ can be bounded as in Theorem 2 by choosing the generator $\mathcal{F} = \mathcal{F}_K$.

*Proof.* Similarly to the previous proof, it suffices to show that $\ell_H^y \in \mathcal{F}_K$ for any $y \in \mathbb{Y} = \{-1, 1\}$ and $h \in \mathfrak{H}_{NN}$ with $\|o\| \prod_{i=1}^{L} \|W\|_2 \leq 1$. This follows directly

| | $\mathfrak{H}$ | $\ell$ | Condition | IPM |
|---|---|---|---|---|
| | $\mathfrak{H}_L$ | $\ell_1$ | $\|w\|_2 \leq 1$ | $d_{\mathcal{F}_K}$ |
| Regression | | $\ell_2$ | $\|w\|_2 \leq \frac{1-M_{\mathbb{Y}}-|b|}{M_{\mathbb{X}}}$ | $d_{\mathcal{F}_{\mathrm{TV}}}$ |
| | $\mathfrak{H}_g$ | $\ell_1$ | $\frac{2M_{\mathbb{X}}}{\sigma^2}\|w\|_1^2 \leq 1$ | $d_{\mathcal{F}_K}$ |
| | $\mathfrak{H}_{\sigma(L)}$ | $\ell_{\log}$ | $\|w\|_2 \leq \frac{\log(e-1)}{M_{\mathbb{X}}}$ | $d_{\mathcal{F}_{\mathrm{TV}}}$ |
| Classification | $\mathfrak{H}_{\mathrm{SVM}}$ | $\ell_H$ | $\|w\|_2 \leq 1$ | $d_{\mathcal{F}_K}$ |
| | $\mathfrak{H}_{\mathrm{NN}}$ | $\ell_H$ | $\|o\|_2 \prod_{i=1}^L \|W\|_2 \leq 1$ | $d_{\mathcal{F}_K}$ |

**Table 1.** The table summarizes the mapping of various learning settings to corresponding IPMs in Theorem 2 under specified conditions on the learnable parameters $w$. The learning tasks are characterized by the hypothesis class (linear $\mathfrak{H}_L$, Gaussian $\mathfrak{H}_g$, logistic $\mathfrak{H}_{\sigma(L)}$, support vector machines $\mathfrak{H}_{\mathrm{SVM}}$, and neural networks $\mathfrak{H}_{\mathrm{NN}}$) and the loss function $\ell$ ($\ell_1$, logistic $\ell_{\log}$, and hinge $\ell_H$). The formal definitions of the hypothesis classes and the losses are provided in Subsection 3.2 and Appendix A.

from the fact that feedforward neural networks with ReLU activation functions are Lipschitz continuous with bounded Lipschitz constant $\|o\| \prod_{i=1}^L \|W\|_2$, see Scaman and Virmaux [20, Proposition 1], and the fact that $\ell_h$ is Lipschitz continuous with Lipschitz constant 1. □

We note that Theorem 4 as well suggests that regularization constraints on the learnable parameters are sufficient to bound the true risk in an AL setting.

Theorem 3 and Theorem 4 are only two examples on how to constraint the hypothesis class for deriving a suitable generalization bound in an AL setting. We note that a variety of other learning settings employing other losses and other hypotheses classes can be considered. We summarize similar results that allow embedding in various generators in Table 1 and refer the reader to the respective proofs in Appendix A. Similar to Theorem 3 and Theorem 4, the complementary results in Table 1 suggest that the regularization constraints on the learnable parameters seem to play a crucial role for the design of query strategies in AL. However, in several cases, the regularization constraints are dependent on bounds on $M_{\mathbb{X}}$ and $M_{\mathbb{Y}}$.

## 4 Conclusion and Outlook

We derived a bound on the generalization error for AL that is based on the IPM as a measure of representativeness. The bound suggests that a query strategy should sample informative samples while maintaining a distribution of the queried samples that is close to the true marginal distribution. This aligns with common practices in AL.

We augmented the bound with a variety of examples that show how to embed different learning settings in various generators. A key insight from these examples is that the regularization constraints on the learnable parameters seem to play a crucial role for a principled design of query strategies. The results of this analysis, summarized in Table 1, can be used to guide the design of query strategies in AL. To this end, the user must first identify which setting in Table 1 matches their scenario. Once identified, the next step is to derive an algorithm that minimizes an empirical estimate of the relevant upper bound, as done, e.g., in the work of Wang and Ye [14].

Additionally, our results can be used to evaluate the quality of ad hoc query strategies in AL. A necessary step towards such an application is to derive upper bounds to the true risk that employ empirical estimates of the IPM, see, e.g., Sriperumbudur et al. [21] for general discussion on empirical estimates of IPMs.

We note that the choice of the IPM as a measure of representativeness is not unique. Other choices of metrics to measure the representativeness of the samples, such as, e.g., $\phi-$divergences, can be considered [22]. We leave this as an open question for future research.

## Acknowledgement

We acknowledge the support by the Deutsche Forschungsgemeinschaft (DFG) within the Research Training Group GRK 2583 "Modeling, Simulation and Optimization of Fluid Dynamic Applications".

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

# A    More Learning Settings

Theorem 3 and Theorem 4 are only two examples on how to constraint the hypothesis class for deriving a suitable generalization bound in an AL setting. In the following we provide the reader with further results, which are also summarized in Table 1.

We start by looking at the Gaussian hypothesis class defined as

$$\mathfrak{H}_g := \left\{ h : \mathbb{X} \to \mathbb{Y} : h(x) = \sum_{i=1}^{n} w_i g(x, t_i) \right\},$$

where $g(x, t_i) = e^{\left( -\frac{\|x - t_i\|^2}{2\sigma^2} \right)}$, for some fixed $\sigma > 0$, and learnable parameters $w = (w_1, \ldots, w_n) \in \mathbb{R}^n$ and $t_i \in \mathbb{X}$ for any $i = 1, \ldots, n$.

**Theorem 5** (Gaussian Hypothesis Classes). Consider a regression problem employing $\mathfrak{H}_g$ and the $\ell_1$-loss. Assume $\mathbb{X}$ to be compact, with bound $M_{\mathbb{X}}$, and $w$ to be such that $\frac{2M_{\mathbb{X}}}{\sigma^2} \|w\|_1 \leq 1$, then the true risk of a hypothesis $h \in \mathfrak{H}_g$ can be bounded as in Theorem 2 by choosing the generator $\mathcal{F} = \mathcal{F}_K$.

*Proof.* Set $\ell_1^y(x) := \ell_1(y, h(x))$ for any $y \in \mathbb{Y}$. Fix $y \in \mathbb{Y}$ and $h \in \mathfrak{H}_g$. By Theorem 2, it suffices to show that $\ell_1^y \in \mathcal{F}_K$. For arbitrary $t_i \in \mathbb{X}$ observe that

$$\left| \frac{\partial}{\partial x} g(x, t_i) \right| = \frac{1}{\sigma^2} e^{\left( -\frac{\|x - t_i\|^2}{2\sigma^2} \right)} \|x - t_i\|_2$$

$$\leq \frac{2M_{\mathbb{X}}}{\sigma^2}.$$

Thus, the function $f(x) = g(x, t_i)$ is Lipschitz continuous on $\mathbb{X}$ with $\|f\|_L \leq \frac{2M_{\mathbb{X}}}{\sigma^2}$. Finally, for any $x_1, x_2 \in \mathbb{X}$ we have

$$|\ell_1^y(x_1) - \ell_1^y(x_2)| = \left| |h(x_1) - y| - |h(x_2) - y| \right|$$

$$\leq \left| \sum_{i=1}^{n} w_i g(x_1, t_i) - \sum_{i=1}^{n} w_i g(x_2, t_i) \right|$$

$$\leq \sum_{i=1}^{n} |w_i| \, |g(x_1, t_i) - g(x_2, t_i)|$$

$$\leq \sum_{i=1}^{n} |w_i| \frac{2M_{\mathbb{X}}}{\sigma^2} \|x_1 - x_2\|_2$$

$$= \|w\|_1 \frac{2M_{\mathbb{X}}}{\sigma^2} \|x_1 - x_2\|_2.$$

Setting $\|w\|_1 \frac{2M_{\mathbb{X}}}{\sigma^2} \leq 1$ implies that $\|\ell_1^y\|_L \leq 1$ and hence $\ell_1 \in \mathcal{F}_K$. $\qquad\square$

In contrast to Theorem 3 and Theorem 4, Theorem 5 requires the input space $\mathbb{X}$ to be compact. Such assumptions are not uncommon in practice. For example, in image classification, the input space is bounded by the pixel values. For example, considering grey images, it is valid to assume that the pixel

values lie in the compact domain $[0, 1]$. In this case, the assumption of Theorem 5 reduces to $\frac{2}{\sigma^2}\|w\|_1 \leq 1$. Such bounded input domains also show up naturally in other applications, such as geographic locations or financial data. Additionally, in practice, one can preprocess the input data to fit within certain bounds. For example, feature scaling or normalization is commonly used to bound the input space to a fixed interval, see [23, 24]. In these scenarios, the constraints are inherent to the domain $\mathbb{X}$, and they effectively regularize the learning process without being formalized as part of the algorithm. Furthermore, such constraints are useful assumptions in the context of kernel methods. For example, assuming the input data lies within a compact set helps in controlling the Rademacher complexity, which provides better generalization bounds, see [16].

This argumentation can also be applied to a-priori constraints on the codomain $\mathbb{Y}$, which we will use in the following result for the linear hypothesis classes.

**Theorem 6** (Linear Hypothesis Classes)**.** Consider a regression problem employing $\mathfrak{H}_L$ and the $\ell_2$-loss. Assume $\mathbb{X}$ and $\mathbb{Y}$ to be compact, with bounds $M_{\mathbb{X}}$ and $M_{\mathbb{Y}}$, and $w$ and $b$ to be such that $\|w\|_2 \leq \frac{1 - M_{\mathbb{Y}} - |b|}{M_{\mathbb{X}}}$, then the true risk of a hypothesis $h \in \mathfrak{H}_L$ can be bounded as in Theorem 2 by choosing the generator $\mathcal{F} = \mathcal{F}_{\mathrm{TV}}$.

*Proof.* Set $\ell_{\mathrm{H}}^y(x) := \ell_2(y, h(x))$ for any $y \in \mathbb{Y}$. Fix $y \in \mathbb{Y}$ and $h \in \mathfrak{H}_L$. By Theorem 2, it suffices to show that $\ell_2^y \in \mathcal{F}_{\mathrm{TV}}$. To this end, it suffices to show that $|y - w^T x - b| \leq 1$. Note that

$$|y - w^T x - b| \leq |y| + \|w\|_2 \|x\|_2 + |b|$$
$$\leq M_{\mathbb{Y}} + \|w\|_2 M_{\mathbb{X}} + |b|,$$

Thus, setting $\|w\|_2 \leq \frac{1 - M_{\mathbb{Y}} - |b|}{M_{\mathbb{X}}}$ we get $\ell_2^y \in \mathcal{F}_{\mathrm{TV}}$. $\qquad\square$

As previously mentioned, restrictions on the domain $\mathbb{X}$ and codomain $\mathbb{Y}$ are used in Theorem 6. Looking at an example of a learning setting, where $\mathbb{X} = [-1, 1]$ and $\mathbb{Y} = [-0.5, 0.5]$ leads to the classical regularization formulation $\|w\|_2 \leq 0.5 - |b|$.

Next, we look at some more binary classification settings. Consider the hypothesis class of logistic linear functions

$$\mathfrak{H}_{\sigma(L)} := \{h : \mathbb{X} \to \mathbb{Y} : h(x) = \sigma(w^T x), w \in \mathbb{R}^n\}$$

with the sigmoid activation function $\sigma(z) = \frac{1}{1+e^{-z}}$ for $z \in \mathbb{R}$ and learnable parameter $w$. This hypothesis class is often used in combination with the logistic loss function $\ell_{\log}(y, h(x)) = -(y \log(h(x)) + (1-y)\log(1-h(x)))$ for $y \in \mathbb{Y}, x \in \mathbb{X}$ and $h \in \mathfrak{H}_{\sigma(L)}$. We set $\mathbb{Y} = \{0, 1\}$ and denote by $e$ the Euler number.

**Theorem 7** (Logistic Hypothesis Classes)**.** Consider a binary classification problem employing $\mathfrak{H}_{\sigma(\mathrm{L})}$ and the logistic loss. Assume $\mathbb{X}$ to be compact, with bound $M_{\mathbb{X}}$, and $w$ to be such that $\|w\|_2 \leq \log(e - 1)M_{\mathbb{X}}$, then the true risk of a hypothesis $h \in \mathfrak{H}_{\sigma(L)}$ can be bounded as in Theorem 2 by choosing the generator $\mathcal{F} = \mathcal{F}_{\mathrm{TV}}$.

*Proof.* Set $\ell_{\log}^y(x) := \ell_{\log}(y, h(x))$ for any $y \in \mathbb{Y}$. Fix $y \in \mathbb{Y}$ and $h \in \mathfrak{H}_{\sigma(L)}$. By Theorem 2, it suffices to show that $\ell_{\log}^y \in \mathcal{F}_{\mathrm{TV}}$. We first consider $y = 1$ and observe that for any $x \in \mathbb{X}$, we have $|\ell_{\log}^y(x)| = \log\left(1 + e^{-w^T x}\right)$. Similarly, for $y = 0$, we observe that $|\ell_{\log}^y(x)| = \log\left(1 + e^{w^T x}\right)$ for any $x \in \mathbb{X}$. Thus, setting $\|w\|_2 \leq \frac{\log(e-1)}{M_{\mathbb{X}}}$ implies $\|\ell_{\log}^y\|_\infty \leq 1$ and hence $\ell_{\log}^y \in \mathcal{F}_{\mathrm{TV}}$. $\qquad\square$

Next we look at the hypothesis class of linear support vector machines (SVM) given by

$$\mathfrak{H}_{\mathrm{SVM}} := \{h : \mathbb{X} \to \mathbb{Y} : h(x) = \mathrm{sign}(w^T x + b)\}$$

with learnable parameters $w \in \mathbb{R}^n$ and $b \in \mathbb{R}$. The primary loss function used in linear SVM is the hinge loss $\ell_H(y, h(x)) := \max(0, 1 - y(w^T x + b))$ for $y \in \mathbb{Y}, x \in \mathbb{X}$ and $h \in \mathfrak{H}_{\mathrm{SVM}}$. We set $\mathbb{Y} = \{-1, 1\}$.

**Theorem 8** (Support Vector Machines)**.** Consider a binary classification problem employing $\mathfrak{H}_{\mathrm{SVM}}$ and the hinge loss. Assume $w$ to be such that $\|w\| \leq 1$, then the true risk of a hypothesis $h \in \mathfrak{H}_{\mathrm{K}}$ can be bounded as in Theorem 2 by choosing the generator $\mathcal{F} = \mathcal{F}_{\mathrm{K}}$.

*Proof.* Set $\ell_{\mathrm{H}}^y(x) := \ell_{\mathrm{H}}(y, h(x))$ for any $y \in \mathbb{Y}$. Fix $y \in \mathbb{Y}$ and $h \in \mathfrak{H}_{\mathrm{SVM}}$. By Theorem 2, it suffices to show that $\ell_{\mathrm{H}}^y \in \mathcal{F}_{\mathrm{K}}$. We note that the function $\ell_H^y(\hat{y}) := \max(0, 1 - y\hat{y})$ is Lipschitz continuous on $\mathbb{Y}$ with Lipschitz constant 1. Additionally, for any $w \in \mathbb{R}^n$ and $b \in \mathbb{R}$ the affine linear function $w^T x + b$ is Lipschitz continuous on $\mathbb{X}$ with Lipschitz constant $\|w\|_2$. Thus, for any $x_1, x_2 \in \mathbb{X}$ we have

$$|\ell_{\mathrm{H}}^y(x_1) - \ell_{\mathrm{H}}^y(x_2)| \leq \left|(w^T x_1 + b) - (w^T x_2 + b)\right|$$
$$\leq \|w\|_2 \|x_1 - x_2\|_2.$$

Setting $\|w\|_2 \leq 1$ implies that $\|\ell_{\mathrm{H}}^y\|_L \leq 1$ and hence $\ell_{\mathrm{H}} \in \mathcal{F}_K$. $\qquad\square$

