# OpenReview forum: "Bounds on the Generalization Error in Active Learning"
_NLDL.org/2025/Conference — NLDL 2025 Poster_

### Official Review · Reviewer_jtnw · 2024-10-06
**IPM based ERM analysis in Active Learning**

**Confidence:** 3

**Summary:**

The paper aims to understand active learning by deriving new upper bounds on the generalization error. It introduces Integral Probability Metrics (IPMs) to measure how well the selected samples represent the overall data distribution. The key idea is that active learning algorithms can perform better by balancing the selection of informative samples with representative ones. The authors connect different active learning scenarios, defined by various loss functions and hypothesis classes, to these theoretical bounds. They also show that using regularization techniques, which limit the complexity of models, helps ensure these bounds are valid.

**Strengths:**

1. The paper extends the Empirical Risk Minimization (ERM) principle to the active learning setting, providing a solid theoretical foundation for designing active learning algorithms.

2. It emphasizes the importance of combining informativeness and representativeness when selecting samples, which aligns with practical observations in active learning.

3. The paper is mathematically sound, with clear proofs and reliance on established concepts like Rademacher complexity and IPMs.

**Weaknesses:**

1. Calculating IPMs can be resource-intensive, especially in high-dimensional spaces, which might limit practical implementation.

2. The conditions needed for the theoretical bounds, such as specific regularization constraints, may be strict or limited in some real-world applications. It might require further clarification.

**Comments on selected Theorems:**
- Overall, please clarify why 1-Lipschtize is sufficient to prove to a broader audience.

**Theorem 1:**
  - The theorem is a well-known result in statistical learning theory.
  - However, the constant $c$ is not explicitly defined. Typically, $c$ depends on the loss function's bound $k$ or other parameters.

**Theorem 2:**
  - The constant $c$ is not defined in the statement (or $C=c?$ from Theorem 1).
  - The condition $\ell^y \in F$ depends on the specific loss function and hypothesis class, and it may not always hold without additional assumptions.
  - Since $d_\mathcal{F}(P_X, P_Q)$ involves IPMs, which may be challenging to compute in practice, mentioning methods for estimating or approximating IPMs could enhance the practical usefulness of the theorem in the active learning.

 **Theorem 6:**
  - Curious if the assumption is valid choice. What happens if the numerator $1 - M_Y - |b|$ could be non-positive if $M_Y + |b| \geq 1$?

**Minor Comments:**
1. Some of the references missed the year. E.g. [5,6,7,14, 18] and so on..

**Justification:**

The paper's theoretical framework is sound. It appropriately extends existing principles to active learning and uses well-established mathematical tools. The assumptions and conditions under which the results hold are clearly stated. By showing that regularization can ensure the validity of the bounds, it effectively connects theoretical findings with practical machine-learning techniques. Overall, the paper's theoretical contributions are solid and provide good insights into active learning.

**Regarding Generalizing Representativeness Measurement with IPMs**:
Earlier studies measured how well samples represent the data using metrics like Maximum Mean Discrepancy (MMD) or the Wasserstein distance. However, these methods worked only under limited conditions. This paper introduces Integral Probability Metrics (IPMs) to measure representativeness. Using different generator classes, IPMs can include various statistical distances, such as Total Variation and Kantorovic distance. This approach makes measuring representativeness more flexible and adaptable.

---

> ### Author Rebuttal · Authors · 2024-10-22
>
> We thank the reviewer for their time and effort in reviewing our manuscript. Below, we provide a detailed response to comments raised by the reviewer.
>
> > the constant $c$ (in Theorem 1) is not explicitly defined. Typically, $c$
> > depends on the loss function's bound
> > or other parameters.
>
> There is a typo here, the constant $c$ is exactly the constant $k$, i.e., the
> upper bound on the loss function. We corrected this typo, see l. 162, l. 257 and l. 261 in the
> revised manuscript.
>
> **comments on Theorem 2**
>
> > The constant is not defined in the statement
> > (or $C=c$ from Theorem 1).
>
> The constant $C$ is defined in Definition 1, see l. 198. For a better
> readability, we recalled this definition before the statement of Theorem 2, see
> l. 249 in the revised manuscript.
>
> > The condition $l^y \in F$ depends on the
> > specific loss function and hypothesis class,
> > and it may not always hold without additional
> > assumptions.
>
> This is indeed correct. In fact, Theorems 3 to 8 all rely on studying the
> assumption $l^y \in F$ under various learning settings, i.e., under various loss
> functions and hypothesis classes. To better clarify this point we added the
> sentence "... additional restrictions on the learning settings need to be imposed...", see l. 309 in the manuscript.
>
> > mentioning methods for estimating or
> > approximating IPMs could enhance the practical
> > usefulness of the theorem in the active learning
>
> A well known result for approximating IPMs is mentioned in the manuscript, see l. 412.
>
> > What happens if the numerator $1-M_Y
> > -|b| $ could be non-positive if $1-M_Y
> > -|b| \geq 1$?
>
> Choosing $1-M_Y -|b|$ to be non-negative is part of the assumption in theorem 6, since we assume  "...$\mathbb{X}$ and $\mathbb{Y}$ to be compact, with bounds $M_{\mathbb{X}}$
>       and $M_{\mathbb{Y}}$, and  $w$ and $b$ to be such that $\|w\|_2\leq \frac{1-M_Y-|b|}{M_X}$...". Therefore if $1-M_Y -|b|$ happens to be non-positive, the theorem is not valid.

---

### Official Review · Reviewer_6img · 2024-10-07
**Interesting math, potentially good information, lackluster communication**

**Confidence:** 3

**Summary:**

This paper presents a theoretically derived upper bound on generalization risk in active learning environments.  The authors provide mathematical context and justification for their upper bound, then provide examples by applying their theorem to some common learning cases.

**Strengths:**

The authors demonstrate a clear understanding of the mathematics of active learning.  The abstract, introduction, and conclusion offer concise and effective summaries of the work.  Mathematical notation is mostly very consistent.  The authors clearly recognize similar work in the field and what differentiates their presented theorem from established literature.

**Weaknesses:**

Several key topics and values are not clearly explained:
- ERM is never defined in the text.
- $Q$ and $X$ are defined with identical properties but are not clearly distinguished from one another.
- By extension, $g$ and $h$ are both defined as mapping $\mathbb{R}^n \rightarrow \mathbb{R}$ (broadly speaking) without clearly distinguishing the difference between them.
- In line 150, the assumption that "$l(y,h(x))\leq k$ for some $k$>0" is stated.  It's unclear at this point in the text if this is meant to be read as "$l(y,h(x))$ is bound by an upper limit of $k>0$" or "$l(y,h(x))$ is positive and finite".  At first, I interpreted it as the former.  This confused me when you introduced "$l_1(y,h(x)) := |y-h(x)|$" in line 307, which is not inherently bound by an upper limit.  Only after, in Theorem 3, do you clarify that you derive a condition on $w$ that does constrain the value of $l_1(y,h(x))$.  But again, this isn't clearly stated in the text.
- The term "generator" is introduced in line 195 without explaining its relevance for the problem setting.
- This may be a result of one of the above points, but it's unclear to me how the derived theorems and conditions (in Table 1) inform the design of the query strategy, as stated in the conclusion.

In summary, my impression is the authors failed to thoroughly explain some key concepts and terminology which hinders the communication of the main topic of the manuscript.

**Justification:**

I am interested in the ideas proposed in this paper, but I think the communication of the ideas needs work.  Some minor revision of the manuscript and/or clarifications from the authors on some key points could go a long way in clarifying the information presented.  As a result, it's possible my understanding of the paper is flawed, even after several thorough readings.

---

> ### Author Rebuttal · Authors · 2024-10-22
>
> We thank the reviewer for their effort in reviewing our manuscript. Below, we
> provide a detailed response to comments raised by the reviewer.
>
> > ERM is never defined in the text
>
>
> The acronym ERM is now introduced in the first appearance of the term "empirical
> risk minimization", see l. 21 in the revised manuscript.
>
>
>  > $Q$ and $X$ are defined with identical properties but are not
>  > clearly distinguished from one another. By extension, $g$ and $h$ are both
>  > defined as mapping $\mathbf{R}^n \to \mathbf{R}$ (broadly speaking) without
>  > clearly distinguishing the difference between them.
>
>  The introduction of $Q$ is essential for modelling the active-learning
>  paradigm. We clarified that in the notation section, see l. 118 in the revised
>  manuscript. The extension of the function $g$ was intended for the mere purpose
>  of defining the expectation operator. This is, indeed, unnecessary. Therefore,
>  we only used the definition of the expectation operator in the definition of Rademacher complexity, see l. 166 in the revised
>  manuscript.
>
> __regarding the assumption "$l(y,h(x))\leq k$"__
>
> The assumption "$l(y, h(x)) \leq k$ for some $k >0$" indeed means that the loss
> function is assumed to be upper bounded for any random variable $z=(x,y)$ and
> any hypothesis function $h$ in our considered hypothesis class. One can, indeed,
> come up with random variables $z=(x,y)$ and hypothesis function $h$ such that
> the $l_1$-loss, defined in l. 320, is unbounded. However, we assume that
> the combination of the random variables $z$ and the hypothesis functions $h \in
> \mathfrak{H}$ always lead to a bounded loss function. This is a fair and
> standard assumption in statistical learning theory, see reference [16, theorem.
> 26.5] in the manuscript. No changes of the manuscript are required.
>
> > The term "generator" is introduced
> > in line 195 without explaining its
> > relevance for the problem setting.
>
> We added a concise introduction of the term generator, see l. 200 in the revised
> manuscript.
>
> > but it's unclear to me how the derived theorems and
> > conditions (in Table 1) inform the design of the query
> > strategy, as stated in the conclusion.
>
> In principle, one can implement query algorithms that minimize empirical
> estimates of the upper bounds derived in our work. Similar results were
> obtained, for example, in references [14] and [16] in the revised manuscript. Here,
> the authors derived an upper bound to the generalization error in specific
> learning settings and specific IPMs. Then, they designed algorithms to minimize
> these upper bounds.
>
> To address the reviewer’s concern more concretely, we have expanded the
> conclusion section (see line 387 in the revised manuscript).
>
> We hope the revisions improve the clarity and communication of work.

---

### Official Review · Reviewer_8tjT · 2024-10-09
**Assessment of Theoretical Contributions and Practical Implications**

**Confidence:** 4

**Summary:**

The paper derives a family of upper bounds on generalization error in the context of active learning (AL). It combines query strategies based on informativeness and representativeness to optimize the selection of data points for labeling. The authors propose using Integral Probability Metrics (IPMs) to quantify representativeness and link various active learning scenarios with corresponding generalization bounds. The main result shows that regularization techniques are sufficient to ensure the validity of these bounds, offering theoretical insights into the design of more effective active learning query algorithms.

**Strengths:**

The framework is mathematically sound, building on established concepts like empirical risk minimization (ERM), Rademacher complexity, and IPMs. The authors rigorously prove that their bounds hold under specific conditions, especially when applying regularization techniques to control hypothesis complexity.The derivations are logically consistent and follow from well-known theorems in statistical learning theory, such as the use of Rademacher complexity to estimate the generalization error. The assumptions and conditions (e.g., Lipschitz continuity, regularization) are standard in the field, ensuring the correctness of the results under the proposed settings.The authors systematically explore various active learning scenarios, linking them to their corresponding upper bounds. This comprehensive approach allows for a nuanced understanding of how different hypothesis classes and loss functions impact generalization error. Additionally, the paper is well-structured, with a logical flow that guides the reader through complex concepts without losing clarity.

**Weaknesses:**

While the paper presents a rigorous mathematical framework, there are potential concerns regarding the assumptions made in the derivations. The reliance on specific conditions for the validity of the upper bounds may limit the applicability of the results. For instance, the assumption that the training data follows a particular distribution could be restrictive in real-world scenarios where data may not conform to these assumptions. Additionally, the paper could benefit from a more detailed discussion on the implications of violating these assumptions and how it might affect the generalization bounds.

The quality of the research is generally high, but there are areas where the depth of analysis could be improved. For example, while the paper discusses the integration of informativeness and representativeness in query strategies, it does not provide a comprehensive exploration of how these strategies can be effectively combined in practice. A more detailed examination of practical implementations or case studies would enhance the quality of the work and provide clearer guidance for practitioners. Furthermore, the paper could benefit from a more thorough comparison with existing methods in active learning, highlighting the advantages and limitations of the proposed approach.

The lack of empirical validation is a notable weakness. While the theoretical framework is robust, the absence of experimental results to support the claims made in the paper raises questions about the practical applicability of the derived bounds. Empirical studies demonstrating the effectiveness of the proposed query strategies in real-world scenarios would significantly enhance the credibility of the findings. The authors should consider including experiments that compare their approach with existing active learning methods to provide a clearer picture of its performance.

**Justification:**

The assessment of the paper "Bounds on the Generalization Error in Active Learning" highlights several areas for improvement. Firstly, the reliance on specific assumptions about data distribution raises concerns about the applicability of the theoretical results in real-world scenarios, necessitating a discussion on these implications. While the theoretical framework is strong, the paper lacks depth in practical guidance for effectively combining informativeness and representativeness in query strategies, and more examples or case studies would enhance its quality. Additionally, certain mathematical notations and concepts could be made more accessible to readers without a strong background in the field, improving clarity. The paper could also better connect its findings to real-world applications in active learning, emphasizing how the derived bounds can be utilized in practice. Lastly, the absence of empirical studies to support the theoretical claims is a significant weakness; conducting experiments to demonstrate the effectiveness of the proposed strategies would strengthen the paper's credibility. Addressing these points would enhance the paper's impact and contribute more effectively to the field of active learning.

---

> ### Author Rebuttal · Authors · 2024-10-22
>
> We thank the reviewer for their time and effort in reviewing our manuscript.
> Below, we provide a detailed response to concerns raised by the reviewer.
>
> > The
> > reliance on specific conditions for the validity of the upper bounds may
> > limit the applicability of the results. For instance, the assumption that the
> > training data follows a particular distribution could be restrictive in
> > real-world scenarios where data may not conform to these assumptions.
>
> We agree that assumptions play a critical role in the validity of theoretical
> results, and discussing these assumptions is indeed of great importance.
>
> We would like to clarify that our work does not impose any assumptions on the
> distribution of the training data, contrary to what the reviewer has suggested.
> The validity of our results does not rely on the training data following a
> particular distribution.
>
> For the learning settings discussed in
> Theorems 3. and 4., the assumptions made coincide with standard regularization
> constraints that are commonly employed in the literature. As such, we believe that an additional detailed discussion on these assumptions is not necessary.
>
> In contrast, Theorems 5, 6, and 7 involve assumptions on the size of
> $M_\mathbb{X}$ and $M_\mathbb{Y}$—the spaces where the random variables $X$ and
> $Y$ take values. These are not assumptions on the distribution of the training
> data, but rather natural constraints that arise in various learning settings.
> For example, in classification problems, the target variable typically has a
> bounded range, i.e., $M_\mathbb{Y} < \infty$. Similarly, normalization
> techniques used in practice often lead to $M_\mathbb{X}$ being effectively
> bounded.
>
> These constraints are commonly employed, such as in kernel-based methods, where bounding $M_\mathbb{X}$ and $M_\mathbb{Y}$ aids in achieving tighter generalization bounds.
>
> To address the reviewer’s concerns, we have added a more detailed discussion of
> each specific assumption in the revised manuscript, see l. 581 to l. 606 and l. 621 to l. 625 in the revised manuscript.
>
>
> > while the paper discusses the integration of informativeness and
> > representativeness in query strategies, it does not provide a comprehensive
> > exploration of how these strategies can be effectively combined in practice.
>
> We recognize the importance of mentioning some diverse methods to combine both
> criteria. We added a discussion on this topic in the introdution (l. 76) and
> conclusion (l. 386).
>
> > The absence of experimental results to support the
> > claims made in the paper raises questions about the practical applicability of
> > the derived bounds. Empirical studies demonstrating the
> > effectiveness of the proposed query strategies in real-world
> > scenarios would
> > significantly enhance the credibility of the findings. The authors should
> > consider including experiments that compare their approach with existing
> > active learning methods to provide a clearer picture of its performance.
>
>  The primary objective of our manuscript is to provide theoretical insights into
>  the design of query algorithms that can be applied across a wide range of
>  learning settings. As such, our focus was not on developing or empirically
>  evaluating a new specific query algorithm that could be directly compared with
>  existing methods in the literature.
>
> That being said, we fully recognize the importance of demonstrating practical applicability and empirical validation. Although we did not include experimental results in this work, we have cited relevant literature where query algorithms, developed based on the theoretical bounds similar to ours, have been empirically evaluated. These studies provide evidence of how theoretical insights can guide the development of effective active learning methods in practice (see references [14] and [15] in the revised manuscript).
>
> We believe these references will offer readers a clearer understanding of how our theoretical results can be translated into practical algorithms and how they perform in empirical settings.

---

### Meta-Review · Area_Chair_mwN2 · 2024-11-01

**Recommendation:** Accept (Poster)
**Confidence:** 3

**Metareview:**

The paper established ERM generalization bounds for active learning, based on Integral Probability Metrics. Some issues were raised by reviewers, mostly related to lack of direct applications of the proposed theoretical framework to concrete algorithms. Yet, the derivation of the proposed bound is meaningful and could lead to improvments in active learning algorithms. I believe it can be of interest for the audience of the conference, and therefore I recommend to accept it in the program.

**Suggested Changes To The Recommendation:**

1: I agree that the recommendation could be moved down

---

### Decision · Program_Chairs · 2024-11-06

**Decision:**

Accept (Poster)

**Comment:**

We recommend a poster presentation given the AC and reviewers recommendations.